# Alcohol consumption and associated risk factors in Burkina Faso: results of a population-based cross-sectional survey

Bruno Bonnechère [1], Sékou Samadoulougou,[2,3] Kadari Cisse [4,5]
Souleymane Tassembedo,[6] Seni Kouanda,[5,7] Fati Kirakoya-Samadoulougou[4]

For numbered affiliations see end of article.

**Correspondence to**
Professor Bruno Bonnechère;
bruno.bonnechere@uhasselt.be

## ABSTRACT

**Objectives** Lifestyle modifiable risk factors are a leading preventable cause of non-communicable diseases, with alcohol consumption among the most important. Studies characterising the prevalence of alcohol consumption in low-income countries are lacking. This study describes the prevalence of different levels of alcohol consumption in Burkina Faso and its associated factors.

**Design** Data from the 2013 Burkina Faso WHO STEPwise Approach to Surveillance survey were analysed. The prevalence of alcohol consumption over the last 30 days was recoded into categories according to WHO recommendations: low, mid or abusive alcohol consumption. Multinomial logistic regression analyses identified factors associated with the different levels of alcohol consumption.

**Setting** Population-based cross-sectional survey in Burkina Faso.

**Participants** 4692 participants of both sexes aged 25–64 years were included in the study.

**Results** In the whole sample, 3559 participants (75.8% (72.5%–78.7%)) were not consuming any alcohol, 614 (12.9% (10.9%–15.3%)) had low alcohol consumption, 399 (8.5% (7.1%–10.1%)) had mid alcohol consumption and 120 (2.7% (2.0%–3.7%)) had abusive consumption. Age was associated with alcohol intake with a gradient effect and older people having a higher level of consumption (adjusted OR (AOR): 2.36, 95% CI (1.59 to 3.51) for low consumption, 2.50 (1.54 to 4.07) for mid consumption and 2.37 (1.01 to 5.92) for abusive consumption in comparison with no consumption). Tobacco consumption was also significantly associated with alcohol intake with a gradient effect, those with higher tobacco consumption being at higher risk of abusive alcohol intake (AOR: 6.08 (2.75 to 13.4) for moderate consumption and 6.58 (1.96 to 22.11) for abusive consumption).

**Conclusion** Our data showed an important burden of alcohol consumption in Burkina Faso, which varied with age and tobacco use. To effectively reduce alcohol consumption in Burkina Faso, comprehensive control and prevention campaigns should consider these associated factors.

## Strengths and limitations of this study

► To the best of our knowledge, this study is the first national representative study on alcohol consumption within the adult population of Burkina Faso.
► This study presents alcohol consumption level and associated risk factors (age and tobacco consumption) in Burkina Faso using a large representative cohort of 4672 participants.
► The main limitation is that alcohol consumption and its associated risk factors were obtained during interviews (memory and social desirability biases).
► Another limitation of this study is that we used data from 2013 (latest results available at large scale) and that the prevalence may have changed due to the recent implementation of alcohol policies to limit the consumption.

diseases (NCDs) such as cardiovascular disease, cancer, chronic respiratory disease and diabetes.[3] Currently the leading cause of ill health in the world, NCDs account for 7 out of 10 deaths worldwide.[4]

Low/middle-income countries are facing the emergence of NCDs.[5] Although some risk factors have been identified such as high blood pressure,[6] lack of physical activity,[7] inadequate diet,[8] etc, there is still a lack of comprehensive data on other modifiable risk factors, in particular tobacco and alcohol consumption,[9] in Burkina Faso.

Recently, clear evidence of increased alcohol consumption and attributable harm in many low/middle-income countries has been highlighted.[10] Furthermore, it is predicted to have a more harmful effect if an effective policy is not adopted.[11] Currently, the prevalence of alcohol intake among the adult African population is about 30%,[12] which is lower compared with the rest of the world where 40% of the world's adult population consumes alcohol and the average consumption per drinker is 17.1 L per year. Interestingly, the prevalence of lifetime abstention, the level of alcohol consumption

## BACKGROUND

The increase in life expectancy[1] combined with unhealthy behaviours and physical inactivity[2] are linked to a rise in non-communicable

and the drinking patterns vary widely across regions of the world.[13] For example, Eastern Europe and Southern Africa had the most detrimental pattern of drinking scores. At the same time, Europe (Eastern and Central) and sub-Saharan Africa (Southern and West) are the most important consumers of alcohol.[13] In Africa, among current drinkers, the prevalence of heavy drinking varied between 7% and 77%, and the prevalence of daily light drinkers varied between 0% and 21%. Overall, drinking patterns varied significantly between and within the examined African countries[14]; more recent studies showed the same trends in South Africa and Ethiopia.[15 16]

One of the major current concerns related to alcohol consumption in Africa is that alcohol companies have targeted Africa as a new market. With the expected increases in the number of potential new alcohol consumers, especially young people and women, the African continent has indeed been identified by the alcohol beverage industry and market researchers as a key area for alcohol market growth.[17 18] It is, therefore, of particular importance to identify subjects the most at risk of alcohol consumption. Currently, two major risk factors have been identified worldwide: gender with male being more at risk[19] and the socioeconomic level where the effect is less clear as the associations between socioeconomic disadvantage and heavier drinking vary depending on country-level income.[20] Pregnant women are particularly more vulnerable due to the fetus's susceptibility.[21–24]

Another important aspect highlighting the need for local studies is that it has been shown that there are significant disparities in alcohol use between Africans (Ghanaians) residing in Europe and Africa, indicating that migration has a significant impact on drinking habits and also implying that alcohol reduction initiatives may need distinct techniques.[25]

There is currently a lack of information about the prevalence of alcohol consumption in Burkina Faso and the associated risk factors. Differences in cultural factors (eg, beliefs and practices) may influence health status, but social, economic and structural determinants of health during people's lifespans appear to be associated with health inequities between ethnic groups as well. Therefore, cultural influences should not be overemphasised as discrete explanatory factors for health inequities.[26]

A first national survey was conducted in 2013 using the WHO STEPwise Approach to Risk Factor Surveillance (STEPS). The STEPS survey is a simple, standardised method for collecting, analysing and disseminating data in WHO member countries. It covered a representative sample of the adult population. The first analyses performed were about the evaluation of cardiovascular risk.[27] In a previous study, we show that tobacco consumption was highly correlated with alcohol consumption in men.[28] Therefore, in this study, we aimed to investigate the prevalence of alcohol consumption and its associated risk factors. Such analyses are needed to drive more efficient prevention campaigns for both alcohol and tobacco consumption.

## METHODS

### Study settings

Burkina Faso is a landlocked country in West Africa of 272 967.47 km². The country is divided into 13 administrative regions and is limited in the North and West by Mali, in the East by Niger, and in the South by Benin, Togo, Ghana, and Côte d'Ivoire.

In 2020, the population was estimated at 21 510 181 inhabitants.[29] The majority of the population (77.30%) lives in rural areas and relies on agriculture and livestock as the main source of income.[30] The median age of the population is 17.9 years old. In 2018, the total fertility rate was 5.2 and the life expectancy at birth was 62.3 years in 2019.[31] Like most low-income countries, Burkina Faso must face the double burden of infectious and chronic diseases. The country is regularly confronted with outbreaks such as measles, cerebrospinal meningitis and malaria,[32] while NCDs constitute a rising public health problem with only limited financial resources allocated.

In Burkina Faso, several sources of alcohol manufacturing exist such as modern brewery and legally imported alcohol, prohibited alcohol mostly imported from neighbouring countries and traditional beer manufacturers. Burkina Faso is a secular country with more than half of the population declared as being of Islamic faith (61.6%), 22.5% Christians and 15.4% traditional religions.[33]

Harmful alcohol use and its adverse events (ie, road traffic accidents, domestic violence) are however important and raising public health concerns in Burkina Faso.[34]

### Study design

This is an analytical cross-sectional study of the WHO STEPS survey conducted in Burkina Faso in 2013. Complete details on the methodology and sampling procedure were described in the WHO STEPS 2013 Burkina report,[35] and in previous studies using this database.[27 28 36] The participants were selected using a three-stage cluster sampling process described in the sampling method.

### Study population

Participants of the study were adults of both sexes aged 25–64 years old who had been living in Burkina Faso for at least 6 months. Exclusion criterion was if people had disabilities hampering their ability to answer the questions (eg, intellectual disabilities, serious mental disorders, cognitive impairment, etc).

### Patient and public involvement

No patients were involved in the study design or analysis of the anonymous data set.

### Sampling

The STEPS survey of Burkina Faso was conducted on a representative sample of 4800 individuals, and the response rate of the STEPS survey was 99.1% in Burkina Faso.[32] After excluding participants with missing information about sampling weight, 4692 individuals were

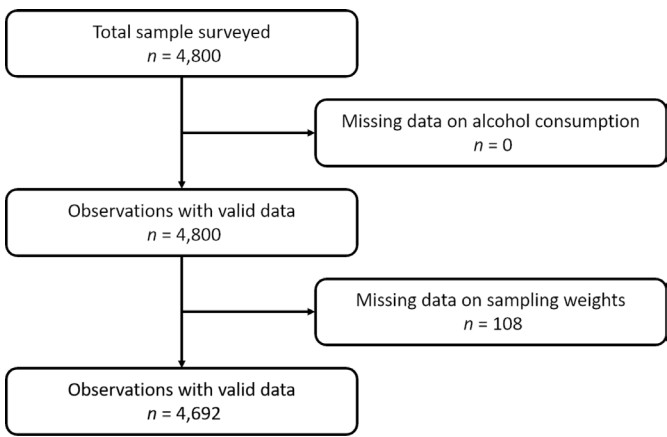

**Figure 1** CONSORT diagram describing study participants, exclusions and analytical sample size. CONSORT, Consolidated Standards of Reporting Trials.

included in the present analysis. The diagram flow of study participants is presented in figure 1. A three-level sampling frame was performed. In the first stage, geographical areas were stratified into rural and urban, andenumeration area (EAs) were selected with probability proportional to their size from both strata. A total of 240 EAs were selected: 185 from rural areas and 55 from urban areas. In the second stage, 20 households were selected from each EA. In the third stage, one person aged 25–64 years in each household was selected using the Kish method.

### Data collection

A full description of the study design and the data collection has been published elsewhere.[27 36] Briefly, all data on alcohol consumption were collected using a standardised questionnaire during face-to-face interviews in the language most spoken by the respondent (step 1 of the STEPS survey). The questionnaires were pretested on the field before being used for the national survey. Alcohol consumption was assessed with the question 'Have you ever consumed any alcohol such as beer, wine, spirits or dolo?'. If the answer is yes, the current

alcohol consumption was assessed with the questions 'During the past 30 days, how many days did you consume alcohol drinks?' and 'When you are drinking, what is the average number of drinks?'. Four levels were defined: no consumption, low, mid and abusive consumption (see table 1 for WHO definition). According to the WHO recommendation, the levels defining the different levels of consumption have been defined differently for male and female.

Other information such as demographic information, anthropometric and behavioural measurements were also collected (step 2). In our study, all the variables used in the manuscript were collected in step 1.

### Study variables

The dependent variable was the mean alcohol consumption in the last 30 days. First, we analysed alcohol consumption to define the profile of the drinker.

The independent variables were: age, sex, education, marital status, occupational status and tobacco consumption (see table 1 for the definition of the variables).

### Data analysis

Some independent quantitative variables were transformed into categorical variables. The list of the variables with the recoded variables is presented in table 1. Categorical variables were described using proportions. The $X^2$ test was used in univariable analysis to test the association between the outcome (the levels of alcool consumption) and the categorical variables (age, sex, living environment, education, marital status, occupational status, tobacco consumption). We first analysed the profile and levels of alcohol consumption using multinomial logistic regression defining no alcohol consumption as the reference value. Then, we identify the profile of the abusive consumer, following WHO recommendations,[37] among current drinkers using logistic regression grouping low and mid consumption as the reference group.

Adjusted ORs (AORs) were calculated for the studied variables and presented with 95% CIs. We then analysed

| Table 1 | Definition of recoded exposure variables |
| --- | --- |
| **Variables** | **Categories** |
| Age groups | "25 to 34 years old", "35 to 44 years old", "45 to 54 years old", "55 to 64 years old" |
| Education | "None", "Primary", "Secondary", "Tertiary" |
| Marital status | "Single", "Married", "Divorced/widowed" |
| Occupational status | "Wage earner", "Self-employed", "Jobless", "Housemaker" |
| Alcohol consumption (WHO recommendations)[37] | None: Never intake of alcohol<br>Low: intake of an average quantity of pure alcohol of less than 40 g per day for men and less than 20 g for women<br>Mid: corresponds to taking an average quantity of pure alcohol of between 40 g and 59.9 g per day for men and between 20 g and 39.9 g for women<br>Abusive: intake of an average quantity of pure alcohol greater than or equal to 60 g per day for men and greater than or equal to 40 g for women. |

Note that one glass of alcohol (beer, wine, 'dolo') contains 10 g of ethanol, the recall period is 30 days.

**Table 2** Sociodemographic characteristics of the study sample and prevalence of alcohol and abusive alcohol consumption

| Variables | Participants, n (%) | Alcohol consumption (95% CI) | | | | P value |
|---|---|---|---|---|---|---|
| | | None | Low | Mid | Abusive | |
| **Age groups, years (n=4692)** | | | | | | |
| 25–34 | 2124 (42.0) | 80.4 (77.1 to 83.4) | 10.6 (8.4 to 13.1) | 7.0 (5.5 to 8.8) | 1.9 (1.2 to 2.9) | <0.001 |
| 35–44 | 1181 (27.9) | 74.8 (70.7 to 78.6) | 13.1 (10.4 to 16.3) | 8.9 (6.8 to 11.5) | 3.1 (1.9 to 4.9) | |
| 45–54 | 841 (18.6) | 71.9 (66.7 to 76.6) | 14.8 (11.0 to 19.6) | 9.5 (7.1 to 12.5) | 3.8 (2.3 to 6.1) | |
| 55–64 | 546 (11.5) | 67.3 (60.4 to 73.5) | 18.3 (14.1 to 23.4) | 11.3 (8.3 to 15.3) | 3.0 (1.5 to 6.2) | |
| **Sex (n=4692)** | | | | | | |
| Female | 2436 (54.3) | 79.6 (75.9 to 82.8) | 8.6 (6.3 to 11.7) | 10.8 (8.8 to 13.3) | 0.9 (0.6 to 1.5) | <0.001 |
| Male | 2256 (45.7) | 71.2 (67.3 to 74.9) | 18.1 (15.5 to 21.1) | 5.8 (4.6 to 7.1) | 4.8 (3.4 to 6.8) | |
| **Living environment (n=4692)** | | | | | | |
| Urban | 945 (24.9) | 72.9 (67.0 to 78.1) | 12.7 (8.4 to 18.6) | 10.0 (7.3 to 13.5) | 4.3 (2.5 to 7.2) | 0.001 |
| Rural | 3747 (75.1) | 76.7 (72.9 to 80.1) | 13.1 (10.8 to 15.6) | 8.0 (6.4 to 9.9) | 2.2 (1.5 to 3.1) | |
| **Education (n=4684)** | | | | | | |
| None | 3622 (77.3) | 77.9 (74.2 to 81.2) | 12.2 (9.9 to 14.8) | 7.8 (6.2 to 9.7) | 2.0 (1.4 to 2.9) | <0.001 |
| Primary | 728 (15.3) | 71.9 (66.8 to 76.6) | 15.6 (11.9 to 20.1) | 8.8 (6.7 to 11.5) | 3.6 (2.0 to 6.3) | |
| Secondary | 334 (7.3) | 60.7 (52.7 to 68.1) | 15.7 (11.7 to 20.7) | 15.3 (11.2 to 20.4) | 8.2 (4.5 to 14.6) | |
| **Marital status (n=4688)** | | | | | | |
| Single | 333 (6.8) | 68.6 (59.5 to 76.4) | 16.9 (11.8 to 23.6) | 9.2 (6.1 to 13.8) | 5.2 (2.6 to 10.1) | <0.001 |
| Married | 4043 (87.2) | 76.5 (73.1 to 79.5) | 12.8 (10.6 to 15.4) | 8.2 (6.7 to 9.9) | 2.4 (1.8 to 3.3) | |
| Divorced/widowed | 311 (5.9) | 73.2 (66.2 to 79.2) | 10.4 (6.9 to 9.9) | 12.5 (8.8 to 17.6) | 3.8 (1.6 to 8.5) | |
| **Occupational status (n=4692)** | | | | | | |
| Wage earner | 281 (6.3) | 62.2 (55.3 to 68.6) | 15.1 (10.7 to 20.9) | 12.7 (8.5 to 18.5) | 9.9 (5.8 to 16.2) | <0.001 |
| Self-employed | 3249 (66.4) | 75.9 (72.2 to 79.3) | 13.2 (11.2 to 15.5) | 7.9 (6.3 to 9.8) | 2.9 (2.1 to 4.3) | |
| Jobless | 125 (2.9) | 77.1 (67.5 to 84.6) | 11.8 (6.7 to 20.0) | 10.4 (5.7 to 18.2) | 0.5 (0.1 to 3.4) | |
| Housemaker | 1037 (24.5) | 78.7 (72.8 to 83.6) | 11.9 (7.6 to 18.2) | 8.9 (6.8 to 11.6) | 0.4 (0.1 to 1.4) | |
| **Tobacco consumption (n=4692)** | | | | | | |
| No | 4217 (90.8) | 77.3 (73.9 to 80.3) | 12.1 (10.0 to 14.6) | 8.4 (6.9 to 10.1) | 2.1 (1.6 to 2.9) | <0.001 |
| <5 cigs/day | 210 (3.8) | 67.7 (59.5 to 74.9) | 23.9 (17.6 to 31.5) | 1.9 (0.8 to 4.5) | 6.4 (3.2 to 12.6) | |
| 5–10 cigs/day | 184 (3.6) | 59.7 (50.8 to 67.9) | 19.8 (13.5 to 28.1) | 13.8 (8.5 to 21.4) | 6.6 (3.2 to 13.0) | |
| >10 cigs/day | 81 (1.7) | 48.1 (32.8 to 63.7) | 18.4 (10.7 to 29.8) | 17.9 (9.8 to 30.1) | 15.5 (4.6 to 40.7) | |

P values indicate the results of the $X^2$ tests.
cigs, cigarettes.

discrepancies between the different regions using AORs for the different regions.

All the analyses were carried out considering the sampling weight and the sampling design. Statistical analyses were performed at an overall significance level of 0.05. Statistics have been conducted in STATA (V.13) and RStudio (V.1.1.442) with R (V.3.4.4).

## RESULTS

Characteristics of the population are presented in table 2. Most of the participants were rural residents (79.7%). The population was predominantly young, with the age group 25–34 years representing 41.9%. Women were more represented, with 54.3% of the total sample.

In the whole population, 3559 (75.8% (72.5%–78.7%)) were not consuming any alcohol, 614 (12.9% (10.9%–15.3%)) had low alcohol consumption, 399 (8.5% (7.1%–10.1%)) had mid alcohol consumption and 120 (2.7% (2.0%–3.7%)) had abusive consumption. There are important differences between the different regions of the countries; the results are presented in figure 2. In the Sahel region, the prevalence is null while in the Sud-Ouest region, the prevalence for all the levels is statistically significantly higher than the country's average values.

The prevalence of abusive consumption was not different in urban setting and rural setting (4.3% (2.5%–7.2%) vs 2.1% (1.5%–3.1%), p=0.19). The repartition of consumption was statistically significantly different between urban and rural environment (p=0.001). There

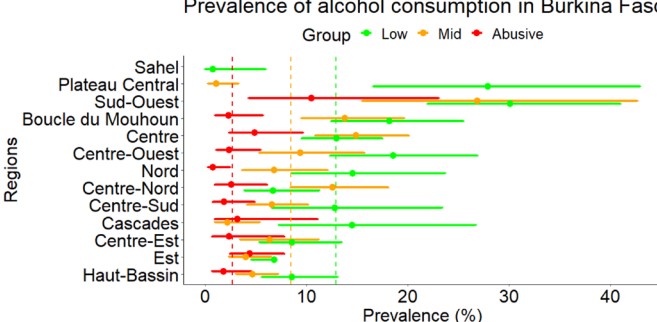

Prevalence of alcohol consumption in Burkina Faso

**Figure 2** Prevalence of the level of alcohol consumption in the different regions of Burkina Faso. Vertical lines indicate the mean prevalence for the whole country.

was a higher proportion of low alcohol consumption in the rural area (13.1% compared with 12.7%); while in the urban area, there is more mid (10.0% vs 8.0%) and high consumption (4.3% vs 2.2%).

In multivariable analysis (table 3), first concerning the alcohol consumption, there is a statistically significant gradient effect of the age regardless of the level of consumption. We found that compared with no alcohol consumption, the odds of engaging in low alcohol consumption were 1.46 times higher among people aged 35–44 years compared with those aged 25–34 years old (AOR: 1.46 (1.09 to 1.95)). The odds were 1.76 times higher (AOR: 1.76 (1.34 to 2.40)), and 2.36 (1.59 to 3.51) for the age groups 45–54 and 55–64 years old, respectively, compared with the age group 25–34 years old. Regarding

**Table 3** Factors associated with alcohol and abusive alcohol consumption in population aged 25–64 years

| Variables | Low consumption | | Mid consumption | | Abusive consumption | |
|---|---|---|---|---|---|---|
| | n | AOR (95% CI) | n | AOR (95% CI) | n | AOR (95% CI) |
| Age groups (years) | | | | | | |
| 25–34 (n=2224) | 221 | Ref | 151 | Ref | 41 | Ref |
| 35–44 (n=1181) | 161 | **1.46 (1.09 to 1.95)\*** | 112 | **1.55 (1.14 to 2.12)\*\*** | 30 | **1.86 (1.06 to 3.27)\*** |
| 45–54 (n=841) | 123 | **1.79 (1.34 to 2.40)\*\*\*** | 81 | **1.81 (1.25 to 2.62)\*\*** | 30 | **2.67 (1.36 to 5.25)\*\*** |
| 55–64 (n=546) | 105 | **2.36 (1.59 to 3.51)\*\*\*** | 55 | **2.50 (1.54 to 4.07)\*\*\*** | 19 | **2.37 (1.01 to 5.92)\*** |
| Sex | | | | | | |
| Female (n=2436) | 204 | Ref | 270 | Ref | 22 | Ref |
| Male (n=2256) | 406 | **2.68 (1.96 to 3.67)\*\*\*** | 129 | **0.38 (0.26 to 0.55)\*\*\*** | 98 | **2.89 (1.56 to 2.37)\*\*** |
| Living environment | | | | | | |
| Urban (n=945) | 111 | Ref | 109 | Ref | 30 | Ref |
| Rural (n=3747) | 499 | 1.18 (0.59 to 2.33) | 290 | 1.12 (0.68 to 1.87) | 90 | 0.76 (0.40 to 1.45) |
| Education | | | | | | |
| None (n=3622) | 452 | Ref | 281 | Ref | 81 | Ref |
| Primary (n=728) | 112 | **1.47 (1.05 to 2.06)\*** | 67 | 1.47 (0.99 to 2.18) | 19 | 1.60 (0.81 to 3.18) |
| Secondary (n=334) | 46 | **2.08 (1.07 to 4.03)\*** | 49 | **2.97 (1.48 to 5.92)\*\*** | 19 | **3.50 (1.34 to 9.15)\*\*** |
| Marital status | | | | | | |
| Single (n=333) | 57 | Ref | 31 | Ref | 17 | Ref |
| Married (n=4043) | 514 | 0.66 (0.40 to 1.10) | 93 | 0.74 (0.43 to 1.30) | 93 | 0.56 (0.23 to 1.36) |
| Divorced/widowed (n=311) | 39 | 0.56 (0.29 to 1.08) | 10 | 0.83 (0.42 to 1.63) | 10 | 1.01 (0.31 to 3.33) |
| Occupational status | | | | | | |
| Wage earner (n=281) | 47 | Ref | 35 | Ref | 20 | Ref |
| Self-employed (n=3249) | 436 | 1.10 (0.58 to 2.06) | 246 | 0.90 (0.53 to 1.51) | 95 | 0.65 (0.26 to 1.67) |
| Jobless (n=125) | 16 | 0.75 (0.37 to 1.53) | 15 | 0.77 (0.33 to 1.76) | 1 | **0.05 (0.01 to 0.44)\*\*** |
| Housemaker (n=1037) | 111 | 2.20 (0.95 to 5.08) | 103 | 0.74 (0.41 to 1.33) | 4 | 0.23 (0.04 to 1.15) |
| Tobacco consumption | | | | | | |
| No (n=4217) | 512 | Ref | 354 | Ref | 86 | Ref |
| <5 cigs/day (n=210) | 47 | **1.72 (1.13 to 2.62)\*** | 6 | 0.52 (0.21 to 1.31) | 14 | **2.42 (1.07 to 5.63)\*** |
| 5–10 cigs/day (n=184) | 36 | 1.60 (0.98 to 2.62) | 23 | **4.20 (2.25 to 7.83)\*\*\*** | 11 | **2.48 (1.06 to 5.79)\*** |
| >10 cigs/day (n=81) | 15 | 1.77 (0.83 to 3.75) | 16 | **6.08 (2.75 to 13.4)\*\*\*** | 9 | **6.58 (1.96 to 22.11)\*\*** |

\*P=0.05, \*\*P=0.01, \*\*\*p<0.001.
AOR, adjusted OR; cigs, cigarettes.

mid consumption, the odds were 1.55 (1.14 to 2.12), 1.81 (1.25 to 2.62), and 2.50 (1.54 to 4.07) times higher for 35–44 years, 45–54 years and 55–64 years, respectively, compared with 25–34 years old. Similar results were found when considering abusive alcohol consumption (AOR: 1.86 (1.06 to 3.27), 2.67 (1.36 to 5.25), 2.37 (1.01 to 5.92)). Interestingly, the effect of gender is different regarding the level of consumption: there is an increased risk in male for low (AOR: 2.68 (1.96 to 3.67)) and abusive consumption (AOR: 2.89 (1.56 to 2.37)), while there is a protective effect against moderate consumption (AOR: 0.38 (0.26 to 0.55)) compared with female. The level of education is also associated with an increased risk, regardless of consumption level (AOR: 2.08 (1.07 to 4.03), 2.97 (1.48 to 5.92), and 3.50 (1.34 to 9.15) for low,

mid, and abusive consumption, respectively, for participants who reached secondary levels in comparison with people without any education).

There is also an important gradient effect associated with tobacco consumption: the consumption of small tobacco quantity (<5 cigarettes/day) is associated with an increased risk of low alcohol consumption (AOR: 1.72 (1.13 to 2.62)), while the consumption of high quantity of tobacco is associated with mid and abusive alcohol consumption (AOR: 6.08 (2.75 to 13.4) and 6.58 (1.96 to 22.11), respectively).

Finally, we computed the risk of having abusive consumption in those consuming alcohol (table 4). Sex is an important risk factor with increased risk for men compared with women (AOR: 2.53 (1.38 to 4.68)) as well

**Table 4** Factors associated with abusive alcohol consumption in population aged 25–64 years consuming alcohol

| | Alcohol consumption | Abusive alcohol consumption | |
|---|---|---|---|
| **Variables** | **N** | **N** | **AOR (95% CI)** |
| Age groups (years) | | | |
| 25–34 | 413 | 41 | Ref |
| 35–44 | 303 | 30 | 1.55 (0.84 to 2.86) |
| 45–54 | 234 | 30 | 1.91 (0.95 to 3.85) |
| 55–64 | 179 | 19 | 0.99 (0.38 to 2.57) |
| Sex | | | |
| Female | 496 | 22 | Ref |
| Male | 633 | 98 | **2.53 (1.38 to 4.68)\*\*** |
| Living environment | | | |
| Urban | 250 | 30 | Ref |
| Rural | 879 | 90 | 0.68 (0.34 to 1.36) |
| Education | | | |
| None | 814 | 81 | Ref |
| Primary | 198 | 19 | 0.89 (0.43 to 1.86) |
| Secondary | 114 | 19 | 1.25 (0.48 to 3.25) |
| Marital status | | | |
| Single | 105 | 17 | Ref |
| Married | 930 | 93 | 0.70 (0.32 to 1.53) |
| Divorced/widowed | 94 | 10 | 1.41 (0.45 to 4.46) |
| Occupational status | | | |
| Wage earner | 102 | 20 | Ref |
| Self-employed | 777 | 95 | 0.51 (0.19 to 1.50) |
| Jobless | 32 | 1 | **0.05 (0.01 to 0.47)\*\*** |
| Housemaker | 218 | 4 | **0.14 (0.02 to 0.78)\*** |
| Tobacco consumption | | | |
| No | 952 | 86 | Ref |
| <5 cigs/day | 67 | 14 | 1.71 (0.68 to 4.26) |
| 5–10 cigs/day | 70 | 11 | 1.20 (0.51 to 2.86) |
| >10 cigs/day | 40 | 9 | 2.26 (0.63 to 8.05) |

\*P=0.05, \*\*p=0.01, \*\*\*p<0.001.
AOR, adjusted OR; cigs, cigarettes.

**Table 5** Risk of alcohol consumption in the different regions of the country

| Region | N | Alcohol consumption | |
| --- | --- | --- | --- |
| | | n | AOR (95% CI) |
| Centre | 548 | 168 | Ref |
| Boucle du Mouhoun | 469 | 147 | 1.23 (0.63 to 2.39) |
| Cascades | 166 | 45 | 0.58 (0.28 to 1.19) |
| Centre-Est | 398 | 72 | 0.51 (0.24 to 1.06) |
| Centre-Nord | 434 | 87 | 0.67 (0.35 to 1.29) |
| Centre-Ouest | 390 | 121 | 0.94 (0.45 to 1.99) |
| Centre-Sud | 217 | 51 | 0.66 (0.28 to 1.54) |
| Est | 376 | 63 | 0.40 (0.22 to 0.74)** |
| Haut-Bassin | 509 | 89 | 0.39 (0.24 to 0.66)*** |
| Nord | 420 | 72 | 0.66 (0.28 to 1.58) |
| Plateau Central | 236 | 62 | 1.01 (0.37 to 2.75) |
| Sahel | 315 | 4 | 0.02 (0.01 to 0.15)*** |
| Sud-Ouest | 214 | 148 | 5.63 (1.98 to 16.01)*** |

Model is adjusted for age, sex, living environment, education, marital status, occupational status and tobacco consumption.
Alcohol consumption includes the different levels of consumption.
*P=0.05, **p=0.01, ***p<0.001.
AOR, adjusted OR.

as the occupational status with jobless people and house-maker associated with a decreased risk of having abusive consumption (AOR: 0.05 (0.01 to 0.47) and AOR: 0.14 (0.02 to 0.78), respectively).

The risk also varied from the different regions. AORs for the different regions of the countries are presented in table 5. We observed that the Sud-Ouest region has a higher risk of consumption compared with the rest of the country. In Sahel, Est and Haut-Bassin, the risks of consumption are significantly lower.

## DISCUSSION

We report here the results of the first nationally representative survey on the prevalence and risk factors for alcohol consumption in Burkina Faso.

The overall prevalence of people with abusive alcohol consumption is 2.7% (2.0%–3.7%). This consumption is lower compared with other African countries[12] and the rest of the world.[13] However, this prevalence might be underestimated because even though Burkina Faso is a secular country, the majority of its inhabitants have a religious faith that prohibits alcohol consumption that could prevent some people from declaring their alcohol intake; this might be particularly true in the Sahel region. Also, alcohol consumption is blamed by society and could also lead to an underdeclaration (positive social perception bias).[38]

We also found differences in the different regions of the country with the highest levels of consumption found in the Sud-Ouest region. Although the results in

a regional level must be interpreted carefully due to the relative small number of participants included, they are of importance for public health. In this region, almost every household produces local beer ('dolo'); therefore, this might explain this observation. The income disparities and the alcohol availability between the different regions could also explain those differences.[39] From a public health perspective, this region also has the highest prevalence of hepatitis B and C.[40] The inhabitants of these regions seem to therefore the hepatotoxic risk of hepatitis and alcohol consumption. We found a null prevalence in the Sahel region, as this is entirely a Muslim region and the desirability bias may be more important compared with other regions.

We also found gender differences with an increased risk in men compared with women for low and abusive alcohol consumption but an increased risk of moderate consumption for women. It is important to specify here that, following the WHO guidelines, the thresholds used to define the different levels of alcohol consumption are different for men and women (33%–50% lower for women compared with men). The risk of abusive consumption in drinkers is more important in men compared with women.

In most countries, the prevalence is higher in men compared with women,[19 41] but when adjusting for multiple factors such as social supports and financial aspects, this effect seems less important. Compared with men, more women are lifetime abstainers, drink less and are less likely to engage in problem drinking. Of note is that women drinking excessively develop more severe medical problems than men. Biological (sex-related) factors, including differences in alcohol pharmacokinetics as well as its effect on brain function and the levels of sex hormones, may contribute to some of those differences.[42] Since pregnant women are particularly vulnerable, we performed a subgroup analysis. A subgroup of 299 out of the 2449 (12.2%) women was pregnant during the survey; among them, 6 (3.9%) reported alcohol consumption, which is statistically lower compared with non-pregnant women (12.6%, p<0.001).

Interestingly, we observed that age was significantly associated with alcohol consumption with a gradient effect of age on alcohol consumption, but it seems that the level of consumption is not influenced by gender as the AORs are relatively similar for the different categories. The influence of age on alcohol consumption is still unclear and not well documented in the literature, except for binge drinking, where young adults are the most at risk.[43] However, this result should be interpreted carefully, considering our study design. This could be the results of preventive campaigns as reflected by a cohort effect. A large Australian study including seven cross-sectional waves showed indeed that male cohorts born between 1965 and 1974 and female cohorts born between 1955 and 1974 reported higher rates of drinking participation (p<0.05), while the most recent cohorts (born in the 1990s) had lower rates of participation (p<0.01).[44]

Concerning the risk of switching to abusive consumption among drinkers, only two factors have been identified: the risk is increased in male (as for the general consumption) and decreased with the occupational status, probably due to financial constraints.

The association with tobacco consumption is probably one of the most important from a public health view because of the comorbidities and the double burden it presents for the population. In a previous study, we identified people most at risk of tobacco consumption in Burkina Faso: tobacco smoking among men was significantly associated with increased age and alcohol consumption. Analysis of risk factors for other tobacco use stratified by gender shows that age, education, residence, and alcohol consumption were significantly associated with consumption for women, age, and alcohol consumption for men.[28]

As recommended by the WHO, in this paper, we presented risk factors associated with abusive alcohol consumption because of the proven negative effects. It is, however, interesting to note that from a medical point of view, limited and reasonable alcohol consumption could have some health-related benefits: for example, the relation between dementia and cognitive disorders is not linear and limited alcohol consumption has a protective role for dementia[45–47] or the protective effect of alcohol on cardiovascular risk, previously known as 'the French paradox'.[48–50] However, these results must be interpreted with caution, especially since it is known that even in low doses, alcohol consumption transiently increases the risk of cardiovascular accidents.[51] Another important point is that considering the prevention side, experiences showed that adopting a too strong position by prohibiting any consumption or behaviour will lead to poor results.[52 53] Considering these two aspects, low alcohol consumption could be considered as acceptable. Alcohol policies that regulate the physical availability of alcohol are associated with lower alcohol consumption in low/middle-income countries.[54] Burkina Faso is at an embryonic stage in this area. Indeed, the country just had its first specifications plan on alcoholic beverage production, importation and selling adopted in 2020.[55] This plan was jointly approved by the Ministry of Trade and the companies involved in such activities. Some additional measurements have been adopted by the government. These include the prohibition of advertisements for tobacco and alcohol products and, since 19 September 2019, the government also banned the production, importation, and marketing of liqueurs and other spirit drinks in plastic bags and bottles of less than 30 cL to limit the availability.[56] This plan needs to be rigorously implemented first and then rapidly incremented to a stronger legislative law to positively impact behaviour and health outcomes.

The main limitation of this study is that alcohol (and tobacco) consumption was obtained during interviews and is therefore dependent on the faith of the participants. There is, therefore, both a risk of memory bias and social desirability, probably more marked during pregnancy. It can thus be estimated that the numbers and prevalence obtained in this survey underestimate the actual consumption. Another potential limitation is that some well-known risk factors for alcohol consumption were not included in the study because data on these variables have not been collected during the STEPS survey. Part of such variables is socioeconomic status. It is a transversal study; therefore, there is a risk of survival bias indicating that older participants with high consumption may die prematurely due to this consumption.[57] The last limitation is that we used data collected in 2013, thus, these data may not represent the actual situation anymore. We have indeed seen that the government has taken measurements to restrict and limit alcohol consumption, therefore the prevalence may be more important than the current situation. However, in this paper, we mainly focus on the risk factor and we do not think that the risk factor has been much modified between 2013 and today. Another STEPS survey was planned for 2020 to have more recent numbers but unfortunately, the survey could not take place due to the COVID-19 pandemic. In the future, the data of this survey could be used to monitor the level of consumption and determine if these measurements are working and if other measures must be taken.

Despite these limitations, given the study design (cluster sampling design) and the sample size, the results of this study can be extended to the whole of Burkina Faso.

## CONCLUSION

In this study, we reported on the burden of alcohol consumption and associated risk factors in a nationally representative sample of adults in Burkina Faso. Our data showed an important burden of alcohol consumption in Burkina Faso. Tobacco consumption is an important modifiable risk factor associated with alcohol consumption. Zone-specific interventions are needed given the higher burden in urban centres and some specific regions such as the Sud-Ouest region. Health policies in Burkina Faso must henceforth account for the control of alcohol and tobacco consumption since there is a strong relationship between those two important risk factors of NCDs. Alcohol policies that regulate the physical availability of alcohol are associated with lower alcohol consumption and should, therefore, be implemented in Burkina Faso to reduce the burden of alcohol consumption.

**Author affiliations**
[1]REVAL Rehabilitation Research Center, Faculty of Rehabilitation Sciences, Hasselt University, Diepenbeek, Belgium
[2]Evaluation Platform on Obesity Prevention, Quebec Heart and Lung Institute Research Center, Quebec city, Quebec, Canada
[3]Centre for Research on Planning and Development (CRAD), Laval University, Quebec city, Quebec, Canada
[4]Centre de Recherche en Epidémiologie, Biostatistiques et Recherche Clinique, Université Libre de Bruxelles-Ecole de santé publique, Brussels, Belgium
[5]Institut de Recherche en Sciences de la Sante, Ouagadougou, Burkina Faso
[6]Département de recherche clinique, Centre Muraz, Bobo-Dioulasso, Burkina Faso
[7]Institut Africain de Santé publique (IASP), Ouagadougou, Burkina Faso

**Acknowledgements** The authors are thankful to all the people involved in the STEPS survey. BB thanks the people who helped him during the analysis of the data in Burkina Faso. This research was supported by the *Fondation Universitaire de Belgique* (Belgian University Foundation).

**Contributors** BB and FK-S conceived the study. BB performed the analysis. BB, SS and KC proposed an early draft of the paper. FK-S, SK and ST made substantial contributions to the conception and design, analysis and interpretation of the data. FK-S, SK and ST contributed significantly to revising the manuscript. All authors read and approved the final manuscript. BB acts as guarantor.

**Funding** BB received a travel grant from the Academie de Recherche et d'Enseignement Supérieur (ARES - https://www.ares-ac.be/fr/) to conduct the data analysis in Burkina Faso (second analysis of the STEPS survey), interpret the data with local epidemiologists and write the manuscript. KC is funded by ARES in the context of a research program for development focused on cardiovascular diseases in Burkina Faso. Award/grant number is not applicable.

**Disclaimer** The funding body had no role in study design, data collection and analysis, decision to publish or preparation of the manuscript.

**Competing interests** None declared.

**Patient and public involvement** Patients and/or the public were not involved in the design, or conduct, or reporting, or dissemination plans of this research.

**Patient consent for publication** Not required.

**Ethics approval** The protocol of the STEPS survey was reviewed and approved by the Ethics Committee for Health Research of the Ministry of Health of Burkina Faso, which gave clearance in accordance with regulations in force (Deliberation No. 2012-12-092 as of 05 December 2012). Written informed consent was systematically sought and obtained from all participants before inclusion in the study. The confidentiality of study participants was fully respected and the analyses performed did not identify any participant. For this study, we obtained authorisation from the General Directorate of Health to reanalyse the data, and the confidentiality of study participants was preserved.

**Provenance and peer review** Not commissioned; externally peer reviewed.

**Data availability statement** Data are available upon reasonable request. The data set of the STEPS survey that was used in this research is available at the Ministry of Health upon request. Other STEPS surveys can be obtained through the WHO (https://www.who.int/ncds/surveillance/steps/riskfactor/en/).

**ORCID iDs**
Bruno Bonnechère http://orcid.org/0000-0002-7729-4700
Kadari Cisse http://orcid.org/0000-0003-0219-0197

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
