## [Reviewer comments · BMJ Open]

ARTICLE DETAILS

TITLE (PROVISIONAL)	Alcohol consumption and associated risk factors in Burkina Faso: results of a population-based cross-sectional survey.
AUTHORS	Bonnechère, Bruno; Samadoulougou, Sékou; Cisse, Kadari; Tassebedo, Souleymane; Kouanda, Seni; Kirakoya-Samadoulougou, Fati

VERSION 1 – REVIEW

REVIEWER	Risi, Megan University of Rhode Island, Psychology
REVIEW RETURNED	14-Nov-2021

GENERAL COMMENTS	Thank you for the privilege of reading this important work. We often have alcohol papers based in the U.S. and other larger western countries. However, it is important to review and understand issues associated with countries not often represented in research. In the paper, you discuss that the prevalence of alcohol use among the African population is lower than other countries. You then go on to reference a WHO report from 2014 and a paper on global burden of disease from 2010 (which referenced data from 2005). Although it is likely that not much has changed in these regions since those reports, can you confirm through more recent references/reports? Around line 30, you mention that pregnant women are at risk because of the vulnerability of the fetus. This is an important statement and sets the stage for your subsequent subgroup analysis mentioned in the discussion. The way that line is written "a group that is particularly at risk because of the vulnerability of the fetus is pregnant women" is a little confusing. Can you reword it to make it clear earlier in the sentence that you are discussing pregnant women? You discuss the questions surrounding alcohol. In the study there was a question about ever lifetime consumption of alcohol, then later questions about past 30 day alcohol use. Is there a breakdown of individuals (Ns would be sufficient) who have never had alcohol and those who have not had alcohol in the last 30 days? In the discussion and in table 1, you mention differences in alcohol severity between men and women outlined by WHO. Can you add that to the method section in text so that it is clear early on that these differences have been accounted for?
--

	I really liked your discussion regarding religious restrictions on alcohol consumption and social desirability. It was a question I had while reading and it was addressed nicely in the discussion
--	---

REVIEWER	Chaturvedi, Himanshu ICMR-National Institute of Medical Statistics, Bio-statistics
REVIEW RETURNED	15-Nov-2021

GENERAL COMMENTS	Manuscript Title: Alcohol consumption and associated risk factors in Burkina Faso: results of a population-based cross-sectional survey The study is important to understand the alcohol consumption levels and the association of risk factors. The following observations need to be clarified.  1. The general profile of the study population is required to understand the nature of the representativeness of the sample. 2. There is a need to provide the sampling methodology of the study in the appropriate section to understand the design. The readers may not like to go to some other place to read the methodology. It should be briefly mentioned. 3. As stated, the multinomial logistic regression analysis was used for analysis. There is a need to explain why? How it was used? The multinomial logistic regression was used if the dependent variable had more than two outcomes. It is better to explain about these methods and what was the reference values of the outcome variable? 4. The tables should be presented according to analysis. In the Table 2, The participants % is not required only N should be given. 5. The results of Chi-square test are given to know the bivariate association of factors with alcohol consumption levels. It should have been given with prevalence of alcohol use. 6. The adjusted odds ratio (AOR) in Table3 presents the adjusted effects of variables i.e. the effects of all other variables are adjusted. It should be interpreted with caution using the appropriate reference level of Alcohol consumption and respective reference of independent variables. 7. There is a need to explain according to the set methodology of data collection i.e. Step-III methods, whether sampling weights were developed and used for analysis? It can be seen from the profile table whether data needs weighting or not? 8. Tables 3 and 4 may be combined as one table and only the adjusted odds ratio for each category should be given. It is not necessary to mention N repeatedly in all cases.
---

REVIEWER	Weerasinghe, Manjula Rajarata University of Sri Lanka , Department of Community Medicine
REVIEW RETURNED	21-Nov-2021

GENERAL COMMENTS	The manuscript is well written in simple language, comprehensive and easy to understand. This will increase the readership upon publication. Country-specific alcohol data and associated risk factors are important in public health point of view. The topic is important and the research design strong and appropriate to the context. Overall, authors have done a great job by analyzing a secondary dataset (8 years old) in a meaningful way.
---

	Below minor comments/suggestions may further improve clarity of specific points.  1. Based on findings, Burkina Faso has a low prevalence of alcohol intake compared to other African countries and the rest of the world. It would be super interesting to discuss individual or societal protective factors (if any). 2. I would like to know from authors 'what will be your next step?' and "based on your findings how you going to influence public health policies in Burkina Faso?'
--	--

VERSION 1 – AUTHOR RESPONSE

Reviewer: 1

Ms. Megan Risi, University of Rhode Island

Comments to the Author:

Thank you for the privilege of reading this important work. We often have alcohol papers based in the U.S. and other larger western countries. However, it is important to review and understand issues associated with countries not often represented in research.

Thank you for your interest in this paper and your comments.

In the paper, you discuss that the prevalence of alcohol use among the African population is lower than other countries. You then go on to reference a WHO report from 2014 and a paper on global burden of disease from 2010 (which referenced data from 2005). Although it is likely that not much has changed in these regions since those reports, can you confirm through more recent references/reports?

This is indeed a very important point, unfortunately, there are no, to our best knowledge, more recent reports comparing different countries (important work still must be done on this topic). But we added three important references highlighting the important difference between countries and differences between African living in Africa or Europe, highlighting the importance of the living environment (Background, p.5-6)

Around line 30, you mention that pregnant women are at risk because of the vulnerability of the fetus. This is an important statement and sets the stage for your subsequent subgroup analysis mentioned in the discussion. The way that line is written "a group that is particularly at risk because of the vulnerability of the fetus is pregnant women" is a little confusing. Can you reword it to make it clear earlier in the sentence that you are discussing pregnant women?

We rephrased this sentence and add another more recent reference (Background, p.6).

You discuss the questions surrounding alcohol. In the study there was a question about ever lifetime consumption of alcohol, then later questions about past 30 day alcohol use. Is there a breakdown of individuals (Ns would be sufficient) who have never had alcohol and those who have not had alcohol in the last 30 days?

Unfortunately, we do not have this information. The majority of the sample (76%) did not consume any alcohol, we could assume that the people who have not consumed any dose of alcohol in the last 30 days were categorized as non-consumer. This may influence the prevalence of consumption but not much the associated risk factors.

In the discussion and in table 1, you mention differences in alcohol severity between men and women outlined by WHO. Can you add that to the method section in text so that it is clear early on that these differences have been accounted for?

We added this information in the method (Data collection, p.9)

I really liked your discussion regarding religious restrictions on alcohol consumption and social desirability. It was a question I had while reading and it was addressed nicely in the discussion

Thank you again for your positive remarks and feedback about this manuscript.

Reviewer: 2

Dr. Himanshu Chaturvedi, ICMR-National Institute of Medical Statistics

Comments to the Author:

Manuscript Title: Alcohol consumption and associated risk factors in Burkina Faso: results of a population-based cross-sectional survey

The study is important to understand the alcohol consumption levels and the association of risk factors. The following observations need to be clarified.

Thank you very much for your relevant comments that help us to improve the quality of this manuscript.

1. The general profile of the study population is required to understand the nature of the representativeness of the sample.

The general profile of the population is presented in the Methods (section study design). The vast majority of the population lived in rural areas (77,3%, 79,7% in our sample), the median age of the population is 17,9 years old (the majority of our sample (42%) are in the 25-34 years group).

2. There is a need to provide the sampling methodology of the study in the appropriate section to understand the design. The readers may not like to go to some other place to read the methodology. It should be briefly mentioned.

We added a section about the sampling in the methods (Sampling, p.9)

3. As stated, the multinomial logistic regression analysis was used for analysis. There is a need to explain why? How it was used? The multinomial logistic regression was used if the dependent variable had more than two outcomes. It is better to explain about these methods and what was the reference values of the outcome variable?

We used multinomial logistic regression in the first part of the analysis to compare the different levels of alcohol consumption (none, low, mid, or abusive consumption), then in the population drinking alcohol we grouped the low and mid consumption and compared the profiles with those of abusive consumption as it is known that low and moderate alcohol consumption may have a protective effect of health (i.e., cardiovascular disease and dementia), also see Point 6.

4. The tables should be presented according to analysis. In the Table 2, The participants % is not required only N should be given.

Thank for this suggestion. We have removed all the % for table 2.

5. The results of Chi-square test are given to know the bivariate association of factors with alcohol consumption levels. It should have been given with prevalence of alcohol use.

We have deleted the sentences related to the comparison between rural residents' characteristics and urban residents' ones. Since we have not provided results by residence stratum in this report. We have presented the prevalence of alcohol use in the second paragraph of the results section (p.11). Thank you for this remark.

6. The adjusted odds ratio (AOR) in Table3 presents the adjusted effects of variables i.e. the effects of all other variables are adjusted. It should be interpreted with caution using the appropriate reference level of Alcohol consumption and respective reference of independent variables.

We have rephrased the interpretation of AOR in the text (Results, p.11-12). In the new sentences, we have added the reference of independent variables and reference of alcohol consumption.

7. There is a need to explain according to the set methodology of data collection i.e. Step-III methods, whether sampling weights were developed and used for analysis? It can be seen from the profile table whether data needs weighting or not?

In our study, all the study variables used in the manuscript were collected in Step I. According to the methodology developed by the WHO, lifestyle data related to the NCDs risk factors (e.g., tobacco consumption, alcohol consumption, physical inactivity and unhealthy diet) are collected via face-to-face interviews using structured questionnaires. In Step II physical measurement of height, and blood pressure was performed and at the last Step (Step III) biological parameters were measured from a whole blood sample. At each step, the sample weight was recalculated to account for the variation of probability of inclusion due to non-response. In this study, we use the Sample weight of Step 1 to weigh all the analyses presented in the manuscript since alcohol consumption was collected at step I. Precisions about the methodology have been added in the manuscript (p.9)

8. Tables 3 and 4 may be combined as one table and only the adjusted odds ratio for each category should be given. It is not necessary to mention N repeatedly in all cases.

In Table 3 we presented the results of multinomial logistic regression, while in Table 4 it's a logistic regression performed to determine if in the population drinking alcohol (regardless of the consumption level), there is a specific profile for abusive consumption or if it's the same profile as when compared with the no-alcohol consumer (lifestyle abstainer).

Reviewer: 3

Dr. Manjula Weerasinghe, Rajarata University of Sri Lanka

Comments to the Author:

The manuscript is well written in simple language, comprehensive and easy to understand. This will increase the readership upon publication.

Country-specific alcohol data and associated risk factors are important in public health point of view. The topic is important and the research design strong and appropriate to the context. Overall, authors have done a great job by analyzing a secondary dataset (8 years old) in a meaningful way.

Thank you very much for your positive comments and feedback about this paper and analysis.

Below minor comments/suggestions may further improve clarity of specific points.

1. Based on findings, Burkina Faso has a low prevalence of alcohol intake compared to other African countries and the rest of the world. It would be super interesting to discuss individual or societal protective factors (if any).

The prevalence may be overestimated due to social desirability bias and the fact that most of the population has a religious faith that prohibits alcohol consumption. This appears now in the discussion (p.13).

2. I would like to know from authors 'what will be your next step?' and "based on your findings how you going to influence public health policies in Burkina Faso?"

The government take measurement in 2019 (banned the production, importation and marketing of liqueurs and other spirit drinks in plastic bags and in bottles of less than 30 centiliters to limit the availability) and the prohibition of advertisements for tobacco and alcohol. Another STEP survey was planned for 2020 to have more recent numbers but unfortunately the survey could not take place due to the COVID-19 pandemic. We will continue to monitor the level of consumption to determine if these measurements are working and if other measures must be taken, this suggestion has been added in the manuscript (p.17).

VERSION 2 – REVIEW

REVIEWER	Risi, Megan University of Rhode Island, Psychology
REVIEW RETURNED	05-Jan-2022

GENERAL COMMENTS	I really like this paper and thing the edits make it stronger. NCDs are defined twice in the background. Delete definition p 6 around line 20 and only use the acronym. In the discussion, p 16 around line 6, the p-value is equal to .001. Just verify that this should be = and not <. Wonderful job!
---

REVIEWER	Chaturvedi, Himanshu ICMR-National Institute of Medical Statistics, Bio-statistics
REVIEW RETURNED	22-Dec-2021

GENERAL COMMENTS	There is much improvement in the presentation of results. However, some minor suggestions may be incorporated. 1. Abstract in the Result section, page 3 and line no.47. Is it "a heavy consumption" or an abusive consumption? A uniform pattern should be followed in the entire text. 2. In Table 2, the Heading of the column shows the % of low consumption, % of Mid consumption and % of abusive consumption. It may indicate low consumption (in %) with 95% CI and so on for others. All three columns should have one heading "alcohol use" as the main heading and three sub-headings i.e. low, mid, and abusive. 3. In Tables 2 & 3; the prevalence of mid consumption of alcohol was almost double among females compared to males whereas it was reversed in the other level of consumption? The same results were noticed for adjusted odds ratio values i.e. reverse between
--

	males and females. It may be clarified and explained. Is it due to sampling error? 4. Instead of Adjusted Odds Ratio, we may call Adjusted Relative Risk (ARR) as Odds Ratio commonly used in the case-control study. 5. In the Methodology, It should be the “Data analysis” section. Remove ‘and Processing’ words. 6. In the Data analysis section, it is stated that the Chi-squared test was used in univariable analysis to test the association between the outcome and the categorical variables; but results are not provided in the table. It should be mentioned in Table 2 for each categorical variable i.e. with chi-square value and its significance at 5%. 7. In the same section, it is mentioned that Multivariable analysis was then performed: we used a logistic regression model fitted using stepwise backward regression modelling, but it is not done or not relevant also as multinomial logistic regression used for the adjusted relative risk of all the dependent variables included in the model such as age, education, sex, and so on. Accordingly, it should be modified. 8. Table 5, the cell frequency of some categories/regions is very low and also the sample size may not be adequate to estimate the prevalence especially for some places/areas. Instead of taking 13 regions separately, it should be combined based on geographical contiguity or similarity i.e. 4 to 5 regions only for comparisons. The outcome of the analysis is not showing much difference or high relative risks. The prevalence with 95% CI may be given in one column before the column of Adjusted Relative risk. 9. The prevalence of alcohol shown by region in the figure (page 39) is not clear and creates confusion. So, it may be removed.
--	---

VERSION 2 – AUTHOR RESPONSE

Reviewer: 1

Ms. Megan Risi, University of Rhode Island

Comments to the Author:

I really like this paper and think the edits make it stronger.

NCDs are defined twice in the background. Delete definition p 6 around line 20 and only use the acronym.

Thank you for your careful reading! We deleted the second definition.

In the discussion, p 16 around line 6, the p-value is equal to .001. Just verify that this should be = and not <.

Indeed we rounded it so we now indicated that is smaller than .001

Wonderful job!

Reviewer: 2

Dr. Himanshu Chaturvedi, ICMR-National Institute of Medical Statistics

Comments to the Author:

There is much improvement in the presentation of results. However, some minor suggestions may be incorporated.

Thank you very much for your comments on this version and on the previous one.

1. Abstract in the Result section, page 3 and line no.47. Is it “a heavy consumption” or an abusive consumption? A uniform pattern should be followed in the entire text.

We modify the term heavy by abusive to be consistent across the manuscript.

2. In Table 2, the Heading of the column shows the % of low consumption, % of Mid consumption and % of abusive consumption. It may indicate low consumption (in %) with 95% CI and so on for others. All three columns should have one heading “alcohol use” as the main heading and three sub-headings i.e. low, mid, and abusive.

Modifications have been done.

3. In Tables 2 & 3; the prevalence of mid consumption of alcohol was almost double among females compared to males whereas it was reversed in the other level of consumption? The same results were noticed for adjusted odds ratio values i.e. reverse between males and females. It may be clarified and explained. Is it due to sampling error?

We don't think that it's due to sampling error but this difference it's interesting to note from a public health point of view. We add the difference in the results (p.12) and discuss further it in the discussion (p.14).

4. Instead of Adjusted Odds Ratio, we may call Adjusted Relative Risk (ARR) as Odds Ratio commonly used in the case-control study.

This is a transversal study, therefore in epidemiological study we present the ORs. And not the RRs.

5. In the Methodology, It should be the “Data analysis” section. Remove ‘and Processing’ words.

Modification has been done.

6. In the Data analysis section, it is stated that the Chi-squared test was used in univariable analysis to test the association between the outcome and the categorical variables; but results are not provided in the table. It should be mentioned in Table 2 for each categorical variable i.e. with chi-square value and its significance at 5%.

We now updated Table 2, we included the prevalence of non-consumer and indicate the results of the chi-squared for each individual variables.

7. In the same section, it is mentioned that Multivariable analysis was then performed: we used a logistic regression model fitted using stepwise backward regression modelling, but it is not done or not relevant also as multinomial logistic regression used for the adjusted relative risk of all the dependent variables included in the model such as age, education, sex, and so on. Accordingly, it should be modified.

You are right we did not perform stepwise regression modelling but multivariable analysis including all the different studied variables, we adapt the text accordingly.

8. Table 5, the cell frequency of some categories/regions is very low and also the sample size may not be adequate to estimate the prevalence especially for some places/areas. Instead of taking 13 regions separately, it should be combined based on geographical contiguity or similarity i.e. 4 to 5 regions only for comparisons. The outcome of the analysis is not showing much difference or high relative risks. The prevalence with 95% CI may be given in one column before the column of Adjusted Relative risk.

Except for the Sahel region (n=4), for the other regions there are enough, in our opinion, participants to estimate the prevalence and the 95%CI. Grouping different regions together would make sense from a statistical point of view, but from a public health perspective it's important to analyze the region independently as specific prevention campaign can be driven at the regional levels. We also indicated that the results of the region comparison, must be interpreted carefully due to the small sample size (p.13).

9. The prevalence of alcohol shown by region in the figure (page 39) is not clear and creates confusion. So, it may be removed.

This is quite a standard way of presenting the data with mean and 95%CI, we think it helps the reader to directly visualize the different level of consumption in the different regions.

VERSION 3 – REVIEW

REVIEWER	Chaturvedi, Himanshu ICMR-National Institute of Medical Statistics, Bio-statistics
REVIEW RETURNED	24-Jan-2022
GENERAL COMMENTS	The manuscript has been revised and may be suitable for publication. Overall, the decision for publication may be taken by the Editors team of BMJ Open,